# Impact of a Targeted Project for Shortening of Imaging Diagnostic Waiting Time in Patients with Suspected Oncological Diseases in Hungary

**DOI:** 10.3390/medicina59010153

**Published:** 2023-01-12

**Authors:** Zsombor Mátyás Papp, László Szakács, Shayan-Salehi Hajivandi, Ildikó Kalina, Edit Varga, Gergely Kiss, Ferenc Solymos, István Takács, Magdolna Dank, Ibolyka Dudás, Tímea Szanka, Csaba László Dózsa, Balázs Rékassy, Béla Merkely, Pál Maurovich-Horvat

**Affiliations:** 1Department of Radiology, Medical Imaging Centre, Semmelweis University, Korányi Sándor u. 2, 1083 Budapest, Hungary; 2Health Services Management Training Centre, Semmelweis University, Kútvölgyi út 2, 1125 Budapest, Hungary; 3Directorate for Core IT Infrastructure and Critical Applications, Semmelweis University, Üllői út 78/b, 1082 Budapest, Hungary; 4Department of Internal Medicine and Oncology, Faculty of Medicine, Semmelweis University, Korányi Sándor u. 2, 1083 Budapest, Hungary; 5Municipality of Budapest, Városház utca 9-11, 1052 Budapest, Hungary; 6Health Sciences Faculty, University of Miskolc, 3515 Miskolc-Egyetemváros, Hungary; 7Heart and Vascular Center, Faculty of Medicine, Semmelweis University, 1085 Budapest, Hungary

**Keywords:** oncology, imaging, computed tomography (CT), magnetic resonance imaging (MRI), waiting time, diagnostic pathway

## Abstract

*Background and Objectives:* Medical imaging is a key element in the clinical workup of patients with suspected oncological disease. In Hungary, due to the high number of patients, waiting lists for Computed Tomography (CT) and Magnetic Resonance Imaging (MRI) were created some years ago. The Municipality of Budapest and Semmelweis University signed a cooperation agreement with an extra budget in 2020 (HBP: Healthy Budapest Program) to reduce the waiting lists for these patients. The aim of our study was to analyze the impact of the first experiences with the HBP. *Material and Methods*: The study database included all the CT/MRI examinations conducted at Semmelweis University with a referral diagnosis of suspected oncological disease within the first 13 months of the HBP (6804 cases). In our retrospective, two-armed, comparative clinical study, different components of the waiting times in the oncology diagnostics pathway were analyzed. Using propensity score matching, we compared the data of the HBP-funded patients (*n* = 450) to those of the patients with regular care provided by the National Health Insurance Fund (NHIF) (*n* = 450). *Results*: In the HBP-funded vs. the NHIF-funded patients, the time interval from the first suspicion of oncological disease to the request for imaging examinations was on average 15.2 days shorter (16.1 vs. 31.3 days), and the mean waiting time for the CT/MRI examination was reduced by 13.0 days (4.2 vs. 17.2 days, respectively). In addition, the imaging medical records were prepared on average 1.7 days faster for the HBP-funded patients than for the NHIF-funded patients (3.4 vs. 5.1 days, respectively). No further shortening of the different time intervals during the subsequent oncology diagnostic pathway (histological investigation and multidisciplinary team decision) or in the starting of specific oncological therapy (surgery, irradiation, and chemotherapy) was observed in the HBP-funded vs. the NHIF-funded patients. We identified a moderately strong negative correlation (r = −0.5736, *p* = 0.0350) between the CT/MR scans requested and the active COVID-19 case rates during the pandemic waves. *Conclusion*: The waiting lists for diagnostic CT/MR imaging can be effectively shortened with a targeted project, but a more comprehensive intervention is needed to shorten the time from the radiological diagnosis, through the decisions of the oncoteam, to the start of the oncological treatment.

## 1. Introduction

Medical imaging is essential for the early diagnosis of oncological diseases. Although diagnostic modalities alone do not improve patient survival, they are key tools in ensuring that patients receive timely and effective treatment [1].

Nowadays, the incidence of oncological diseases is increasing in developed countries, which is partly attributed to the increase in life expectancy. According to the Hungarian Central Statistical Office, the life expectancy in Hungary was 68.2 years for men and 76.5 years for women in 2001, and it increased to 70.7 years and 77.5 years by 2021, respectively [2]. This fact alone shows the increased need for modern imaging techniques; however, allocating the limited healthcare resources is currently a major challenge.

A marker that can be used to assess the overall success of the diagnostic methods for any oncological disease (whose progression is well known) is the length of diagnostic workup measured in days (time to diagnosis), as this has a significant impact on the patient’s therapeutic options, i.e., the chance of survival [3]. Several studies have confirmed that the use of Computed Tomography (CT) and Magnetic Resonance Imaging (MRI) as early diagnostic tools is not only cost-effective, but also increases quality-adjusted life years (QALYs) and potentially increases survival rates [4,5,6,7,8]. Therefore, the time spent on diagnostic imaging investigations and the total diagnostic interval, from the first suspicion or detection of the disease to the initiation of oncological therapy, are considered as important indicators. The detection of cancer on time, the limitations of the waiting time of the entire diagnostic period, and the lengths of the period in general from the first suspicion of cancer to treatment are highlighted as key efficiency indicators in several international health policy papers and guidelines [9,10].

In Hungary, the priority given to oncological care was highlighted by an official regulation years ago. According to the official statement, from 1 July 2015 onwards, in the case of a well-established clinical suspicion of a malignant tumor, the provider of CT/MRI medical care must perform the necessary diagnostic imaging examination within 14 days of the date of referral, with no special deadline defined for creating the medical report based on the imaging examination. This care is financed by the National Health Insurance Fund (NHIF) within the framework of the social health insurance system. However, the providers are unable to accept new patients after the NHIF-funded capacity is exhausted, even if they had capacities for additional examinations; hence, the imaging procedures proved to be less available in everyday clinical practice in the timeframe of 14 days, leading to the creation of waiting lists—this happened with much greater frequency during the COVID-19 pandemic.

To eliminate/reduce the waiting lists for the diagnostic imaging for patients with suspected malignant diseases, the Municipality of Budapest and Semmelweis University signed a cooperation agreement with the public (NHIF-funded) and private diagnostic imaging providers as a part of the framework of the Healthy Budapest Program (HBP) in October 2020. According to this cooperation agreement, CT/MRI imaging investigations should be performed within 7 days, and the final imaging medical reports should be prepared within a further 3 days in patients with suspected oncological disease. The HBP purchases cancer-related imaging diagnostic services at the current NHIF funding rate, which is provided with an additional budget to cover the costs of these further imaging investigations for patients living in specific districts of Budapest. These contracted capacities are performed in a separate time window and therefore do not impede the timely treatment of the NHIF-funded cases. On the contrary, those who are enrolled in the HBP even shorten the appointment times for NHIF-funded cases. The HBP not only aimed at shortening the waiting lists for diagnostic imaging, but also facilitated the necessary follow-up tests and treatment for the patients already diagnosed. It could also be expected that the experience gained might further improve the healthcare system in the capital and speed up patient pathways, particularly in the field of oncological care.

The aim of the current study was to evaluate the first experiences with the HBP, focusing on the waiting times for diagnostic procedures in the above-mentioned indications. Our research hypothesis stated that the time from referral to CT/MRI imaging would be shorter for patients in HBP compared to the patients with regular care provided by NHIF. Therefore, we analyzed the date between the HBP-funded and the NHIF-funded patients regarding the time intervals from the first suspicion of an oncological disease to the delivering of the final CT/MRI medical reports. In addition, the subsequent diagnostic time intervals of the patients’ pathways after the CT/MRI investigations were also analyzed.

## 2. Materials and Methods

In our retrospective, two-armed, comparative clinical investigation, HBP-funded and NHIF-funded imaging radiology procedures with CT/MRI scans were analyzed, with particular attention to the time course of initiation, performance, and reporting of the scans.

The data for the current analysis were extracted from the Semmelweis University’s hospital information system and the Medical Imaging Centre’s image archive system. Our database includes patients who participated in a CT/MRI examination at Semmelweis University between 16 November 2020 and 31 December 2021. In this database, investigations financed by the NHIF were marked as “reimbursement category 1”, while the reimbursement category for patients within HBP was marked as “BP”.

We included patients in the analysis if they were examined for the first time due to suspected oncological disease. Patients were excluded from the study if they were already included in our database with a given pathology or a suspected pathology from 1 January 2015. We used ICD-10 (International Classification of Diseases, 10th version) codes for the diagnosis of cancers. We did not include patients in the analysis with incomplete basic clinical data (gender and age) or with obvious administrative errors. Further exclusion criteria were as follows: address outside Budapest, examination findings were not finalized, referral diagnosis could not be categorized, and the dates of the request and examination were the same. 

We compared HBP-funded vs. NHIF-funded imaging pathways. We used propensity score matching to achieve the best possible comparability between the two arms. The propensity score matching was performed using the R program, with a 1:1 matching ratio. 

We compared the HBP and the NHIF cohorts with regard to every identifiable step of the imaging process. We could analyze the time intervals between 

The first mentioning (first suspicion) of an oncological disease and the request for an imaging examination;Submitting (referring) the imaging request to the Medical Imaging Centre and performing CT/MRI scans;Performing CT/MRI scan and preparing/delivering the final radiological report.

In the first analysis, we assessed how long it took for the referring physician to request the imaging examination from the time the diagnosis was mentioned. Here, we looked only at the new patients—who had no previous imaging with oncological indications—with an internal referral since we only had the available anamnestic data in our medical system of that subset of patients (*n* = 92 vs. *n* = 92). 

The second and the third analyses were performed on the entire cohort (*n* = 450 vs. *n* = 450), but the sum of these two analyses (total time interval between submitting the imaging request and delivering the final medical report) were also reported. Notably, the second and third analyses were performed in the entire cohort first, followed by a repeated analysis, but with the exclusion of 26.9% of the NHIF-funded patients where the dates of the imaging request and the investigation were identical. 

We also investigated the patients’ pathways after having the imaging procedures completed. In this respect, we analyzed the time needed for the different steps of the histological investigations. In addition, the elapsed time before receiving a decision from the multidisciplinary team (“oncoteam”) was also analyzed. Notably, various cancer types have different diagnostic pathways. These can be divided into two categories: (a) when the tumor is described and the disease can be clearly defined (e.g., by cytology and core biopsy and (b) when staging imaging is required before a therapeutic decision can be made. The multidisciplinary team can only give a therapeutic plan if the tumor diagnosis and staging are available. Finally, we analyzed the time interval between the decision of the multidisciplinary cancer team and the start of oncological treatment (surgery, chemotherapy, and irradiation). 

In another sub-analysis, we examined the proportion of referral pathology confirmed by imaging; for this analysis, we analyzed the data from 100 HBP-funded and 100 NHIF-funded patients selected at random. 

Finally, since the coronavirus disease 2019 (COVID-19) pandemic occurred during the period of the study, we examined its impact on the imaging performance. In March 2020, the Hungarian government ordered a lockdown due to COVID-19. Although this was lifted later, some restrictions remained, resulting in difficulties for the patients in accessing health care providers. In our analysis, we investigated whether the COVID-19 pandemic waves (assessed with the numbers of new COVID-19 cases, as registered in the NHIF central database) had any impact on the numbers of requests for CT/MRI investigations.

The data analysis was conducted using the statistical software R. Mean values with standard deviation (mean ± SD) are given. The bootstrap method-based 95% confidence intervals are also reported. The Fisher exact test was used for assessing the justification of the radiological investigations, while the Spearman correlation was investigated to evaluate the relationship between COVID-19 cases and radiological requests. A value of *p* < 0.05 was considered statistically significant.

The study was conducted according to the guidelines of the Declaration of Helsinki. The study design was ethically approved by the Medical Research Council, Scientific and Research Committee, Budapest, Hungary (code number: IV/3298-1/2022/EKU, date of the approval: 27 April 2022). The data protection met all requirements prescribed by General Data Protection Regulation (GDPR).

## 3. Results

### 3.1. Basic Characteristics of Patients

During the 13 months of the study period, a total of 20,566 CT/MRI scans were performed at the Semmelweis University Medical Imaging Centre (average scan: 1.74/patient, min: 1, max 15 scans); out of these, 6804 cases had a referral diagnosis of oncological disease (definitive or suspected). The CT/MRI examinations were HBP-funded in 801 cases and NHIF-funded in 6003 cases. Considering the basic exclusion criteria, the HBP-funded cohort had 450 patients, while the number of patients in the NHIF-funded cohort was 1876. The propensity score matching method was used, resulting in our final HBP-funded and NHIF-funded cohorts (450 and 450 patients in both groups) (Figure 1). 

The basic characteristics of the HBP-funded patients (*n* = 450) and those of the NHIF-funded patients (*n* = 450) were as follows: proportion of females: 58% and 54%; age of patients: 60.3 ± 18.7 years and 63.6 ± 14.4 years; the proportion of patients with a new diagnosis of definitive or suspected cancer: 66% and 59%; the proportion of patients with “internal” referral for imaging: 64% and 73%, respectively. In our HBP-funded and NHIF-funded cohorts, the number of patients with CT scans was higher than those with MRI scans (CT: *n* = 285 and *n* = 342; MRI: *n* = 165 and *n* = 108; in HBP-funded and NHIF-funded patients, respectively).

### 3.2. Participation in the HBP by Districts of Budapest

We could observe great differences between the various districts of Budapest regarding the number of referred patients and the share of HBP-funded patients (*n* = 801). The highest numbers of referred and the second highest proportion of HBP-funded patients were from the 8th district, where Medical Imaging Centre is located (Figure 2). Importantly, none of the districts exhausted the maximal HBP-funded investigation limit, based on the targeted HBP support.

### 3.3. Time Intervals for Steps of Imaging Process

The time interval between the first mentioning (first suspicion) of oncological disease and the request for imaging examination could be investigated only in the new patients with an “internal” referral. A significantly shorter time interval was found in the HBP-funded (*n* = 92) vs. the NHIF-funded (*n* = 92) groups (mean: 16.1 vs. 31.3 days, respectively, mean difference: −15.2 (95% CI: −25.1 to −5.3) days; *p* = 0.005).

In the entire matched cohort, the time interval between submitting the imaging request and the imaging examination was significantly shorter in the HBP-funded (*n* = 450) vs. the NHIF-funded (*n* = 450) patients (mean 3.1 vs. 10.8 days, respectively, mean difference: −7.8 (95% CI: −9.2 to −6.6) days; *p* < 0.0001). The time interval between imaging investigation and the delivering of the final medical report was shorter in the HBP-funded vs. the NHIF-funded patients (3.2 vs. 4.5 days, respectively, mean difference: −1.3 (95% CI −1.9 to −0,6; *p* <0.0001). Taken together, the total time interval between the request for imaging and the delivering of the final report was significantly shorter in the HBP-funded vs. the NHIF-funded patients (6.3 vs. 15.3 days, respectively, mean difference −9.0 (95% CI −10.7 to −7.8; *p* < 0.0001) (Table 1).

In the entire matched cohort, but excluding 26.9% of the NHIF-funded patients where the dates of the imaging request and imaging investigation were the same, similar but more pronounced differences were found between the HBP-funded (*n* = 450) and the NHIF-funded (*n* = 329) patients regarding the time intervals for the steps of the CT/MRI imaging process (Table 2).

Taken together, in the HBP-funded vs. the NHIF-funded patients, the time interval from the first suspicion of oncological disease to the request for imaging examinations was on average 15.2 days shorter (16.1 vs. 31.3 days), and the mean waiting time for the CT/MRI examination could be significantly reduced by 13.0 days (4.2 vs. 17.2 days, respectively). In addition, the imaging medical records were prepared on average 1.7 days faster for the HBP-funded patients than for the NHIF-funded patients (3.4 vs. 5.1 days, respectively).

### 3.4. Time Intervals for Histological Examinations

We identified those patients whose histological examinations, after the imaging process, took place at Semmelweis University. There was no significant difference between the HBP-funded and the NHIF-funded patients regarding the time interval from histological sampling to sample receipt at pathology (1.8 and 1.5 days, respectively, mean difference 0,3 (95% CI: −0.5 to 1.4) days, *p* = 0.4828). Accordingly, no significant difference was observed between the HBP-funded and the NHIF-funded patients regarding the time interval from sample receipt to final, validated histological reports (3.5 vs. 2.6 days, respectively, mean difference 0,9 (95% CI: −0,3 to 2.1) days, *p* = 0.2031).

Taken together, we found that HBP did not have a significant effect on the speed of the histological examinations.

### 3.5. Time for the Decision of the Multidisciplinary Oncological Team

We found that in both the HBP-funded and the NHIF-funded groups the oncological teams reviewed the patients within 21 days, as established in the university’s policy. Nevertheless, there was no significant difference between the HBP-funded and the NHIF-funded patients regarding the elapsed time needed for the decision of the multidisciplinary team (12.4 and 15.9 days, respectively, mean difference −3.5 (95% CI: −9.1 to 1.4) days, *p* = 0.2024).

### 3.6. Time Interval between the Multidisciplinary Team Decision and the Start of the Therapy

There was no significant difference between the HBP-funded and the NHIF-funded patients regarding the interval between the multidisciplinary cancer team decision and the start of therapy (37.9 and 34.5 days, respectively, mean difference 3.4 (95% CI −10.6 to 17.4) days, *p* = 0.6571). 

### 3.7. Justification of Radiological Investigations

It was found that the suspicion of a tumor was confirmed by the requested imaging examination in 56.0% and 62.0% of cases in the HBP-funded and NHIF-funded groups, respectively (*p* = 0.4668). Regarding the final result of the diagnostics, there was no detected difference between the HBP-funded and the NHIF-funded groups, which means that the patient selection and referral in the frame of the HBP was not distorted or selected by other aspects.

### 3.8. Impact of COVID-19 Pandemic on Imaging Scans

We identified a moderately strong negative correlation between the monthly numbers of cases with CT/MRI requests during the examined period (*n* = 20,566) and the new COVID-19 cases in Budapest (*n* = 192,277) during the pandemic waves (r = −0.5736, *p* = 0.0350) (Figure 3). 

## 4. Discussion

Our study showed that the CT/MRI imaging of the patients with cancer or with suspected cancer was performed in a significantly shorter time under HBP, which is a type of targeted funding support with an extra budget, as compared to the general care based on NHIF funding. Nevertheless, no further shortening of different time intervals during the subsequent oncology diagnostic pathway or in the starting of specific oncological therapy was observed in the HBP-funded vs. the NHIF-funded patients after having the medical report of the CT or MRI investigations.

Medical imaging is essential for the early detection of different oncological diseases and for determining treatment plans; therefore, shorter examination times are important in cancer outcomes [11]. The HBP provided extra financial support to shorten the waiting time for imaging for patients with suspected malignancies. The successful reduction in diagnostic imaging waiting time should lead to a further shortening of the entire diagnostic period, resulting in earlier therapeutic decisions. Unfortunately, there was no decrease in the subsequent diagnostic time intervals after the CT/MRI investigations, but this aspect has not yet been included in the HBP. In the case of the potential extension of the project, the entire diagnostic period should be additionally supported and followed.

Our study showed that there were significant differences in the number of patient referrals between the districts of the capital, while none of the districts exhausted the previously determined, available framework. The program was launched during the COVID-19 pandemic, and we suspected that the disruption it caused might have provided an explanation for the lower referral numbers than previously expected. Our analysis exposed a moderately strong correlation between the number of referrals and the number of new COVID-19 cases. Our research experience showed that the main problem was the willingness of the out-patient care polyclinic sites to refer cases, as only a small number of patients were enrolled in the program, while the imaging diagnostic side was able to meet its commitments and significantly reduce the previously usual diagnostic time. To ensure the success of the program, it is necessary to increase the number of patients enrolled, which can be achieved by, for example, expanding the number of institutions and providers eligible for referral and motivating them. It is also necessary to examine in depth what other factors have contributed to the low referral rates. It is possible that the providers have not received sufficient information or assistance in carrying out the administrative tasks. This could be helped by an oncology patient pathway management system [12]. Financial compensations for referring institutions or a reduction in the administrative burden on doctors and specialists would certainly help to increase the uptake of HBP [13]. 

The territorial coverage obligation of the Budapest healthcare institutions is quite complicated, and the different departments of the university were not the designated care providers for all patients; ultimately, this circumstance limited our research scope. Nevertheless, there was the possibility that the patients who did not present for follow-up or who, based on the data from the university’s medical system, did not appear to have presented before a multidisciplinary team, may have received care at another institution. In addition, the COVID-19 pandemic also introduced changes in patient pathways, as it did in other countries involved with this pandemic [14,15,16,17]. 

While HBP is a major step forward in accelerating oncological diagnostics, it could only affect the initial part of oncological care. In the case of cancer or suspected cancer, the primary goal of care is not only early diagnosis, but also the timely initiation of effective therapy. This implies that it is worthwhile to consider ways of optimizing the oncological pathway at other stages of the patient’s journey to improve patient care. The success of the oncology patient’s journey cannot be achieved in the way that it has been conducted so far, it needs to be scaled up: case management activities need to be encouraged and comprehensive patient journey management needs to be implemented to track and guide the patient’s journey through the entire diagnostic and therapeutic pathway.

Our findings should also be assessed in terms of the limitations of the analysis. The study was a retrospective, single-center, comparative analysis. Although the single center has advantages in terms of uniformity, it is a limitation in terms of generalizability. Being able to access only the documentation and data generated during the care events at Semmelweis University was a limitation of our study. The two study groups showed some differences at the baseline, but by using propensity score matching, the patients included in the analysis were well comparable.

## 5. Conclusions

In the 13 months covered by the analysis, a total of 801 imaging examinations were carried out under the HBP program at Semmelweis University, Budapest. Overall, our study has shown that there is potential to reduce the time needed to perform advanced CT/MRI imaging examinations, which are essential in the investigation of cancer or suspected cancer. Nevertheless, more comprehensive intervention is needed to shorten the subsequent time intervals from the radiological diagnosis to the start of the appropriate oncology treatment.

## Figures and Tables

**Figure 1 medicina-59-00153-f001:**
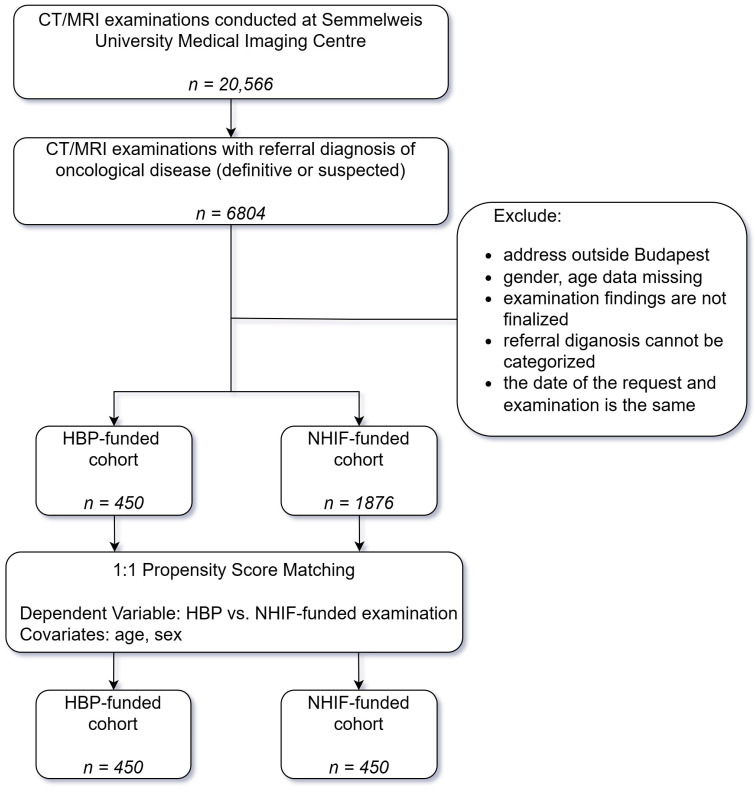
Flowchart of the investigation. HBP: Healthy Budapest Program, NHIF: National Health Insurance Fund.

**Figure 2 medicina-59-00153-f002:**
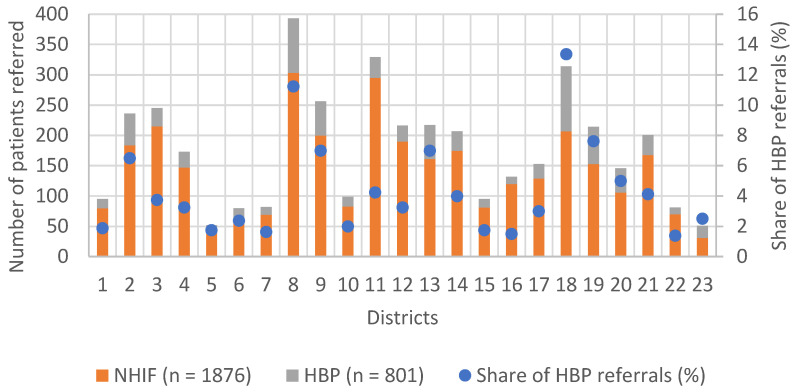
Share of patient referrals by 23 districts in Budapest.

**Figure 3 medicina-59-00153-f003:**
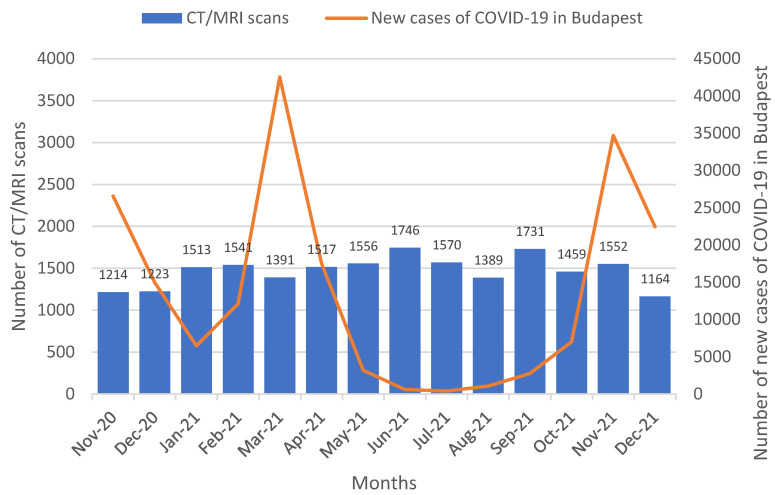
Monthly numbers of CT/MRI scans performed at Semmelweis University, Medical Imaging Centre during COVID-19 pandemic in Hungary from November 2020 to December 2021. Orange line: COVID-19 pandemic waves in Hungary.

**Table 1 medicina-59-00153-t001:** Time intervals for steps of CT/MRI imaging process in HBP-funded vs. NHIF-funded patients with definitive or suspected diagnosis of malignancies (entire cohort).

	HBP-Funded Patients (*n* = 450)	NHIF-Funded Patients (*n* = 450)	Mean Difference (95% CI)	Significance *p* Value
Time interval (day) between imaging request and investigation	3.1	10.8	−7.8 (−9.2; −6.6)	<0.0001
Time interval (day) between investigation and delivering medical report	3.2	4.5	−1.3 (−1.9; −0.6)	<0.0001
Total time interval (day) between imaging request and delivering medical report	6.3	15.3	−9.0 (−10.7; −7.8)	<0.0001

HBP: Healthy Budapest Program, NHIF: National Health Insurance Fund.

**Table 2 medicina-59-00153-t002:** Time intervals for steps of CT/MRI imaging process in HBP-funded vs. NHIF-funded patients with definitive or suspected diagnosis of malignancies (entire cohort, but excluding 26.9% of NHIF-funded patients where dates of imaging request and investigation were identical).

	HBP-Funded Patients (*n* = 450)	NHIF-Funded Patients (*n* = 329)	Mean Difference (95% CI)	Significance *p* Value
Time interval (day) between imaging request and investigation	4.2	17.2	−13.0 (−14.4; −11.6)	<0.0001
Time interval (day) between investigation and delivering medical report	3.4	5.1	−1.7 (−2.5; −1.0)	<0.0001
Total time interval (day) between imaging request and delivering medical report	7.6	22.3	−14.7 (−16.4; −13.1)	<0.0001

HBP: Healthy Budapest Program, NHIF: National Health Insurance Fund.

## Data Availability

The datasets generated and/or analyzed during the current study are available from Z.M.P. on reasonable request.

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
