# Peer review of "Impact of a Targeted Project for Shortening of Imaging Diagnostic Waiting Time in Patients with Suspected Oncological Diseases in Hungary"

_medicina, 2023, doi:10.3390/medicina59010153_

Round 1

Reviewer 1 Report

The aim of this study was to evaluate an extra founded program and its ability to shorten CT/MRI waiting time for suspected cancer patients in comparison to the NHIF.

Shortening the time from clinical suspicion to starting the treatment is essential for patient outcome and programs made to reduce this waiting time are very important.

 It is logical that higher budget program will be able to accelerate these diagnostic procedures.

I think that the authors should try to explain in detail the way their program worked (did they contracted more private clinics to be able to reduce waiting time? Paid more for same procedure comparing to the NHIF?  Not all clinics are in contract with the NHIF? NHIF nr of examinations is limited? Ex X examination/month/clinic). Please provide more detailed information regarding HBP strategy.

How the HBP patients where prioritized?   

Author Response

Dear Reviewer,

We appreciate the time and effort you dedicated to providing feedback on our manuscript and are grateful for the insightful comments on our paper.

Thank you for pointing out that some details of the inner workings of the Hungarian healthcare system might not be evident to the reader.

The rationale for the Program was that when providers were unable to accept new patients after the NHIF-funded capacity was exhausted, there was a marked increase in the time it took to book high-value diagnostic imaging tests.

The Municipality of Budapest, which is responsible for the Program, has sought to contract sufficient public (NHIF contracted) and private (providing paid services outside the health insurance) CT and MRI diagnostic capacity in the capital (partly through the voluntary entry and partly through successful public procurement) to avoid queues and to meet the target of 7 + 3 days for examination and diagnostic time. These contracted capacities do not impede the timely examination of NHIF-funded cases, with scans taking place in separate time slots in addition to existing capacities. On the contrary, those who are enrolled in the HBP even shorten the waiting time for NHIF-funded cases.

The HBP's strategy is to speed up the time till the initial investigation after the first suspicion of cancer. Further investigations and the vast majority of CT and MR scans required for recurrent therapies are covered by the NHIF.

As suggested, we have included additional information in the Introduction section (lines 80-98) that should clarify the HBP strategy.

Reviewer 2 Report

This manuscript presented a study comparing the imaging diagnostic wait times between HBP (Healthy Budapest Program)-funded and NHIF (National Health Insurance Fund)-funded cases in Hungary. The outcomes demonstrated that HBP achieved its target and shortened the waiting time for cancer-related patients.

I have three concerns about this manuscript:

1   1. The strategy of the HBP to decrease the waiting time was not well discussed. According to the author’s description, the HBP included an agreement that “CT/MRI imaging investigations should be performed within 7 days, and the final imaging medical reports should be prepared within a further 3 days in patients with suspected oncological disease”, and the program had the budget to support the fast diagnostic result. Is it the reason for the shorter waiting time that patients in HBP acquired higher diagnostic priority due to the program framework? If this is the case, will the limitation of resources (e.g. number of imaging devices) be an issue when involving a large scale of patients?

2   2. The discussion about COVID-19 is barely related to the aim/conclusion of the study.

3   3. There are some minor writing issues:

The sentence between lines 28-29 "The study database included 6804 cases of CT/MRI examinations within the 28 first 13 months of the HBP" is confusing. Please clarify the 6804 cases included both HBP-funded and NHIF-funded cohorts

Figure 1 and its discussion between lines 177-184 are not well-matched. For example, the 1876 cases of NHIF-funded cohort after basic exclusion were not mentioned in Figure 1, while the number of 3266 in figure 1 was not discussed.

Figure 2 did show the number of HBP-funded patients in each district, which was indicated in the legend.

For the x-axis labels of Figure 3, please use the month number or year+3 letter month abbreviations to avoid confusion.

Author Response

Dear Reviewer,

We appreciate the time and effort you dedicated to providing feedback on our manuscript and are grateful for the insightful comments on our paper.

Point 1: The strategy of the HBP to decrease the waiting time was not well discussed. According to the author’s description, the HBP included an agreement that “CT/MRI imaging investigations should be performed within 7 days, and the final imaging medical reports should be prepared within a further 3 days in patients with suspected oncological disease”, and the program had the budget to support the fast diagnostic result. Is it the reason for the shorter waiting time that patients in HBP acquired higher diagnostic priority due to the program framework? If this is the case, will the limitation of resources (e.g., number of imaging devices) be an issue when involving a large scale of patients?

Response 1:

Thank you for pointing out that some details of the inner workings of the Hungarian healthcare system might not be evident to the reader.

The rationale for the Program was that when providers were unable to accept new patients after the NHIF-funded capacity was exhausted, there was a marked increase in the time it took to book high-value diagnostic imaging tests.

The Municipality of Budapest, which is responsible for the Program, has sought to contract sufficient public (NHIF contracted) and private (providing paid services outside the health insurance) CT and MRI diagnostic capacity in the capital (partly through the voluntary entry and partly through successful public procurement) to avoid queues and to meet the target of 7 + 3 days for examination and diagnostic time. These contracted capacities do not impede the timely examination of NHIF-funded cases, with scans taking place in separate time slots in addition to existing capacities. On the contrary, those who are enrolled in the HBP even shorten the waiting time for NHIF-funded cases.

The HBP's strategy is to speed up the time till the initial investigation after the first suspicion of cancer. Further investigations and the vast majority of CT and MR scans required for recurrent therapies are covered by the NHIF.

As suggested, we have included additional information in the Introduction section (lines 80-98) that should clarify the HBP strategy.

Point 2: 2. The discussion about COVID-19 is barely related to the aim/conclusion of the study.

Response 2:

When we started evaluating the results of the Program, we discovered that significantly fewer examinations had been performed than previously anticipated. We have experienced that the COVID-19 pandemic had a profound effect on the functioning of the Hungarian healthcare system, and we suspected that it might have influenced the implementation of a new program. We are aware that the pandemic caused similar disruptions in other countries as well, therefore, we not only wanted to examine its effect on our Program but at the same time provide an insight into one of the public health effects of COVID-19 in Hungary. We feel that not including this information could be interpreted by the readers as not considering a potentially important factor. Therefore, we chose to incorporate a section in the Discussion (lines 307-312) that explains the relevance of the effect Covid-19 had on our investigation.

Point 3: There are some minor writing issues:

The sentence between lines 28-29 "The study database included 6804 cases of CT/MRI examinations within the 28 first 13 months of the HBP" is confusing. Please clarify the 6804 cases included both HBP-funded and NHIF-funded cohorts

Figure 1 and its discussion between lines 177-184 are not well-matched. For example, the 1876 cases of NHIF-funded cohort after basic exclusion were not mentioned in Figure 1, while the number of 3266 in figure 1 was not discussed.

Figure 2 did show the number of HBP-funded patients in each district, which was indicated in the legend.

For the x-axis labels of Figure 3, please use the month number or year+3 letter month abbreviations to avoid confusion.

Response 3:

Thank you for pointing these out.

We rewrote the sentence in question: “The study database included all CT/MRI examinations conducted at Semmelweis University with referral diagnosis of suspected oncological disease within the first 13 months of the HBP (6804 cases).”

We have modified Figure 1 to contain the most important steps of the exclusion process and matched it to the discussion.

We have corrected Figure 2, now it contains both the number of NHIF and HPB-funded patients in each district.

We have changed the x-axis labels of Figure 3.